# Milk Thistle (*Silybum Marianum* L.) as a Novel Multipurpose Crop for Agriculture in Marginal Environments: A Review

Roberto Marceddu, Lucia Dinolfo, Alessandra Carrubba *, Mauro Sarno * and Giuseppe Di Miceli

Department of Agricultural, Food and Forest Sciences (D/SAAF), University of Palermo, Viale delle Scienze Build, 4 Entr. L, 90128 Palermo, Italy; roberto.marceddu@unipa.it (R.M.); lucia.dinolfo@unipa.it (L.D.); giuseppe.dimiceli@unipa.it (G.D.M.)
* Correspondence: alessandra.carrubba@unipa.it (A.C.); mauro.sarno@unipa.it (M.S.); Tel.: +39-091-23862208 (A.C.); +39-091-23862222 (M.S.)

**Abstract:** Milk thistle (*Silybum marianum* (L.) Gaertn.) is a versatile crop that has adapted to the broadly different soil and environmental conditions throughout all continents. To date, the fruits ("seeds") of the plant are the only reliable source of silymarin, which, given its recognized therapeutic effects and its many present and potential uses, has led to a significant re-discovery and enhancement of the crop in recent years. Overall, although many studies have been carried out globally on the bioactivity, phytochemistry, and genetics of milk thistle, few and discontinuous research activity has been conducted on its basic agronomy as well as on the farm opportunities offered by the cultivation of this species. However, the multiple potential uses of the plant and its reduced need for external inputs suggest that milk thistle can perfectly fit among the most interesting alternative crops, even for marginal environments. The growing interest in natural medicine, the increasing popularity of herbal dietary supplements, and the multiple possibilities for livestock feeding are all arguments supporting the idea that in many rural areas, this crop could represent a significant tool for enhancing and stabilizing farm income. However, several issues still have to be addressed. The species retains some morphological and physiological traits belonging to non-domesticated plants, which make the application of some common agronomic practices challenging. Furthermore, the lack of reliable field data devoted to the definition of suitable cropping protocols represents a major constraint on the spread of this crop among farmers. This review has therefore focused on updating information on the main morphological and phytochemical traits of the crop and its agronomic characteristics and novel uses. Several gaps in technical knowledge have been addressed, and further goals for experimental activity have been outlined in order to guide farmers eager to cope with the cultivation of such a challenging and resource-rich crop.

**Keywords:** alternative crops; bioactive compounds; low-input management; milk thistle; silymarin

## 1. Introduction

Milk thistle (*Silybum marianum* (L.) Gaertn.) is a spiny herb belonging to the *Asteraceae* family. When growing in wild conditions, the plant spends its first season after seed germination at the vegetative stage, therefore being commonly classified as a biennial [1]. Under cultivation, it is mostly grown as an annual crop, with varying cycle durations according to the sowing time [2,3]. The plant, originally grown in Southern Europe and Asia, is now found throughout the world [4]. Milk thistle has been used for medicinal purposes for over 2000 years, most commonly for the treatment of liver disease (cirrhosis and hepatitis) as well as for the protection of the liver from toxic substances [5–8]. The therapeutic effects of milk thistle are closely connected to the presence of a flavonoid complex called silymarin, composed of a mixture of silybin A and B, isosilybin A and B, silychristin, and silydianin [9,10]. The highest amount of silymarin is present within the achenes (improperly but commonly termed "seeds") of the plant [11–13]; however,

the whole plant is also used for medicinal purposes to treat kidney, spleen, liver, and gallbladder diseases [12,14,15].

In the last decade, research into the use of silymarin expanded considerably, embracing the possibility of curing other illnesses and ailments. In addition to its hepato-protective effect [16], silymarin also showed antioxidant, anti-inflammatory, and antifibrotic properties [17]. It was found to stimulate protein biosynthesis, to increase lactation, and to possess immune-modulatory activity [18]. Furthermore, silymarin inhibits cell growth, DNA synthesis, and other mitogenic signals in human prostate, breast, and cervical carcinoma cells [19,20].

It was introduced as a crop in most areas of Europe, Asia, North and South America, and Southern Australia [21,22]. In Poland, which is an important European producer of milk thistle seeds and its derivates, the cultivated area covers about 2000 ha [23]. Its commercial cultivation has recently become more significant in North America [24], where milk thistle is among the top-selling herbal dietary supplements, with retail sales amounting to USD 2.6 million in the mainstream multioutlet channel in 2015 [25]. Likewise, in Italy, milk thistle stands as one of the major cultivated or cultivable medicinal species, ranking fourth for volume used (1,920,000 kg/year), and fifth for its estimated wholesale commercial value (EUR 3,494,400 per year) [26].

In Mediterranean environments, exploratory studies agree in considering milk thistle as one of the most interesting alternative crops, even for marginal environments [3,27,28].

However, in some areas, the plant is considered a noxious weed and treated consequently. In Pakistan, wheat yield losses caused by *S. marianum* infestation ranged from 7% to 37% [29]. In pastures, it is considered a dangerous, invasive species since its large rosette (up to 1 m in diameter) has the potential to displace most other pasture species, and its spiny thistles can hinder the movement and grazing activity of livestock. Additionally, due to the plant's ability to accumulate nitrogen, some cases of livestock intoxication after milk thistle ingestion have been reported, especially when the ingested plant was in the early wilting stage [29–31].

Moreover, several of the plant's features, including its spiny habit, fruit shattering, and asynchronous ripening, represent a constraint to the spread of its cultivation and therefore need to be addressed. This review aims to provide an overview of the major morphological and phytochemical characteristics of milk thistle, as well as discuss the present and potential uses and agronomic traits of the plant in view of its extended field cultivation.

## 2. Origin and Distribution

Milk thistle is native to the Mediterranean basin, within a large area spanning from southern Europe to Asia Minor and northern Africa, although it is also naturalized in other continents [14,32–34]. It is a typical species of the Mediterranean-Turanic chorotype [32]. In Italy, it is spread all over the country, between 0 and 1100 m asl, with the exception of Friuli, most of the Po Valley, and the Alps [1].

Milk thistle has long been known as a useful species. Indeed, archeological surveys have demonstrated its utilization since the Neolithic Era in the Mediterranean environment [35]. Seeds of milk thistle have been used for medicinal purposes for more than 2000 years, mostly to treat liver diseases [36]. Theophrastus (4th century BC) was probably the first one to describe it, under the name "Pternix"; later, the plant was mentioned by Dioscorides in his "*Materia Medica*" (1st century AD) and by Plinius the Elder (1st century AD).

The plant is now widespread throughout the world [37], both in wild populations [38] and as a crop [21], usually cultivated to extract silymarin.

In natural conditions, milk thistle, commonly referred to as a ruderal species, can be found in fertile, highly disturbed environments (e.g., pastures or sheep camps endowed with high soil nitrogen levels), but also in anthropized sites such as roadsides [30]. From natural stands, it spreads easily to cultivated fields, facilitated by its remarkable seed production, easy wind dispersal, and viability. Seeds buried in the soil can remain viable

for 3–4 years [39], or even up to 9 years [31]. Hence, plants of S. *marianum* can be very aggressive and competitive with crops in many cultivated areas, and the species is reported to be a noxious weed [30] on arable land, both in warmer climates (where temperatures rarely fall below 0 °C) [29,40] and in colder regions [41].

### 3. Genetics and Breeding

The genus *Silybum (Asteraceae)* includes two species: *S. marianum* (L.) Gaertn. and *S. eburneum* Coss. and Durieu [42,43]. Hetz et al. [44] argued that the two forms are probably only variants of the same species due to the easy cross-pollination and interfertility occurring between the two genotypes. The same Authors also reported that in crossing experiments between *S. marianum* and *S. eburneum,* the number of fruits produced was relatively high as compared to the two parental species. *S. marianum* is a diploid species (2n = 34) [42], and even though its flowers are often visited by pollinating insects, it was described to be autogamous, with an average outcrossing rate of 2% under field conditions [44].

Despite the increasing interest in this species as a multipurpose crop and its actual economic importance in the herbal market, the plant has not been subjected to a thorough breeding activity so far. Limited genetic research has been carried out only since the early 2000s, mainly to develop high-yielding cultivars with elevated silymarin content [45]. Hence, with the lack of specific breeding that addresses agronomical issues, cultivated plants are still in possession of several plant traits that are typical of undomesticated species, including fruit dispersal at maturity (Figure 1), asynchronous flowering, spiny leaves, and erratic outcomes of yield quality and stability. These features are not uncommon in Medicinal and Aromatic Plants (MAPs) [46], and in many cases, they are supposed to be developed through the plant's evolutionary process to ensure the best reproductive success and environmental fitting for them. However, they represent a severe constraint on proper agronomical practice, and a straightforward breeding activity needs to be carried out.

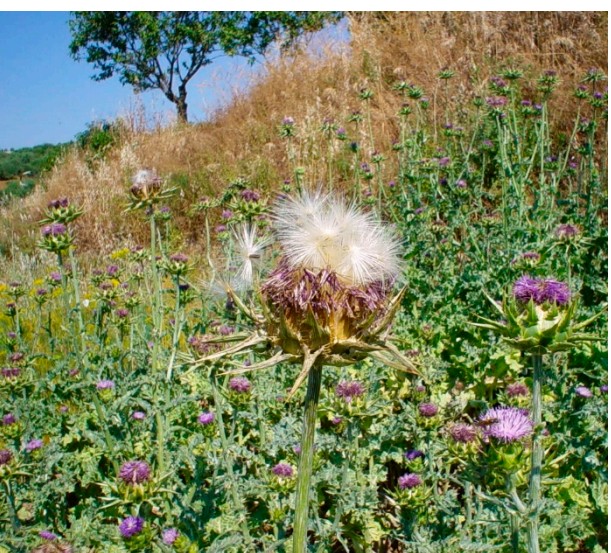

**Figure 1.** Fruit shattering at maturity of *Silybum marianum* (Photo: A. Carrubba).

The identification of key genes involved in fruit shattering and the identification of shatter-resistant genotypes were performed by Martinelli [47], who elucidated the physiological basis of shattering and found it to be controlled by air relative humidity, with a substantial influence of the fruit linkage to the receptacle and the conformation of the fruit pappus. In line with this research outcome, this author identified three vigorous and stable lines as well as discussed the consequent modifications of biomass composition, plant structure, and habit [47].

Asynchronous flowering is the other major problem with the milk thistle crop. Indeed, at the time of harvest, the plants bear flower heads at all stages of development, with inherent problems for proper crop management, including mechanized harvest [33]. To cope with this issue, the cultivar "Argintiu" was developed in the Republic of Moldova, characterized by simultaneous seed maturation in flower heads [48]. Furthermore, breeding efforts have been addressed in order to obtain plants with a reduced number of secondary flower heads [3].

The issue of the milk thistle's spiny habit was addressed in Pakistan in the late 1980s, but the attempt to obtain spineless mutants using radiation is still ongoing [49].

Other goals of breeding are concerned with the qualitative aspect of fruits, especially the silymarin content. The discovery of genes involved in the biosynthesis and accumulation of silymarin in fruits integument during development [50–53], or genes implicated in the synthesis of other valuable constituents of the fruits [45], are interesting starting points in this direction. So far, only a few improved cultivars for silymarin production have been registered such as the Polish variety "Silma" [54]. The path to genetic improvement and the breeding of milk thistle is still open, and there is room for the selection of more suitable, productive, and metabolite-rich milk thistle cultivars. Wild populations include genotypes that are supposed to represent the maximum expression of this capability to adapt to environmental conditions [3], hence constituting a valuable gene pool for the further exploitation of this species. A thorough investigation of the amount of variability in wild genotypes, obtained from different geographical regions, is therefore recommended [55].

## 4. Description

### 4.1. Morphology

Large variability exists in the morphological traits of *S. marianum*, and a comprehensive descriptor list for the species was compiled in 2016 [56].

The plant has a glabrous or slightly downy stem that is erect and branched in the upper part [57]. Depending on different soil fertility and environmental conditions, individual stem height can vary between 40 and 200 cm.

The basal leaves are alternate, large, and glabrous with spiny margins (Figure 2A). The size of the leaves usually ranges between 50 and 60 cm in length and 20–30 cm in width [37]. Additionally, white veins along the top page of the leaf represent a distinctive feature of the species [3], although the presence of individuals bearing evenly green leaves has been reported [58]. The stem leaves are smaller than those of the rosette.

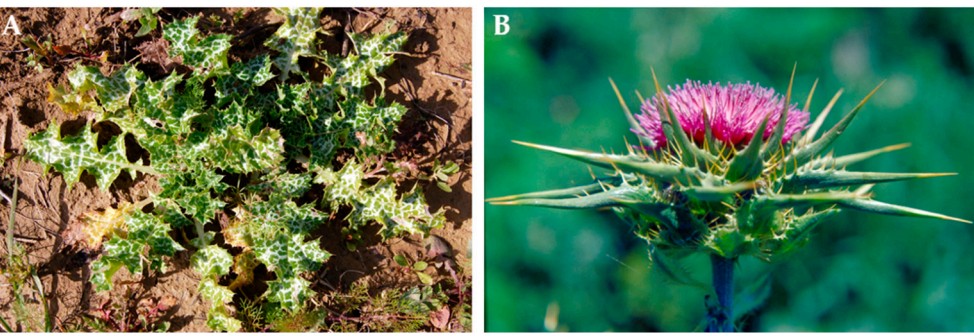

**Figure 2.** Morphological traits of *Silybum marianum* leaves (**A**) and a red purple flower head with spiny bracts (**B**). (Photos: G. Di Miceli).

Each stem, including those from lateral branches, ends with a flower head about 5 cm in diameter [57], elevated above the leaves. The flower is usually red-purple, but a white-flowered genotype was retrieved and investigated by Szilágyi and Tétényi at the beginning of the 1970s [59]; the occurrence of white-flowered genotypes has also been reported in Israel [60].

The inflorescence heads are surrounded by spiny bracts (Figure 2B). The numerous florets are hermaphrodite, with a tubular five-lobed corolla.

The fruits ("seeds") are achenes, characterized by a long white pappus; their color ranges from black to glossy brown, but gray ones with spots are also common [61]. The weight of 1000 seeds varies between 20 and 30 g [23,28]. According to genotype and growth conditions, each flower head can produce from around 65–100 to about 190 seeds, adding up to more than 6000 seeds per plant, of which 94% are viable [28,62].

*4.2. Biology and Physiology*

Milk thistle is generally classified as an annual species even though it can be biennial [1,2,32]. In wild conditions, the seeds germinate in autumn, and flowering occurs in the following summer, reaching a vegetative cycle of 8–9 months long [32].

Just as in many other crops, milk thistle's phenological stages can also be described using a two or three-digit BBCH ("*Biologische Bundesanstalt, Bundessortenamt und CHemische Industrie*") coding system, scaled from 0 (seed germination) to 9 (plant senescence) [63]. Accordingly, the plant's growth pattern can be divided into four stages: germination (BBCH stage 0), growth period (BBCH stages 1–4), flowering (BBCH stages 5–6), and seed development to maturity (BBCH stages 7–8) [37].

The first stage includes seed germination until the emergence of cotyledons through the soil surface (BBCH scale 0.9; 00.9 according to the three-digit scale). Emergence begins from 1 to 3 weeks after sowing, mainly depending on the temperature [63].

The second stage embraces vegetative growth, from the full development of cotyledons (BBCH stage 1.0) to the maximum development of plant biomass (BBCH stage 4.9), including the rosette stage (BBCH stage 3.0). During this stage, the stem remains compressed and close to the soil surface; the plant develops a large rosette and overwinters in this form [37]. At this stage, *S. marianum* is easily identifiable, thanks to its characteristic spiny and variegated leaves, and can compete strongly with neighboring annual plant species [64].

The third stage comprises flowering, pollination, and fertilization. Inflorescences start forming from the central main stem, initially being enclosed between leaves (BBCH stage 5.0), and later, clearly visible between them (BBCH stage 5.1). At the onset of flowering, the stem elongates, producing an erect flower stem [37]. Meanwhile, primary and secondary branches also start to develop at the leaf axil in a basipetal manner. In each head, florets start opening (BBCH stages: 6.0–6.9). Flowering occurs from April to May. The height of the plants and branching attitude are deeply influenced by weather conditions, soil fertility, and sowing density [63].

The fourth stage includes seed growth and filling (BBCH stages: 7.1–7.9). The seeds are ripe in July. During fruit ripening, plant senescence also begins. Usually, once fruits on the side branches have ripened, the entire plant appears completely dead and dry [63].

## 5. Management Techniques

*5.1. Adaptation*

Milk thistle is a long-day plant [37] and is suited to more or less heavy soils with a pH between 5.5 and 7.6 [37].

It is generally considered resistant to extreme temperatures. As for cold tolerance, Martinelli [47] defined the plant as being hardy to Zone 8b, i.e., areas dealing with the lowest daily minimum temperature on a yearly basis, ranging from −9.4 °C to −6.7 °C [43]. The highest cold resistance occurs when the crop is at the rosette stage, which represents the phenological stage in which the species usually overwinters [37].

On the other hand, several plant features such as the spiny habit and the deep root biomass suggest that the crop is also fairly adapted to warm and semi-arid conditions. A few preliminary experiments in Mediterranean areas [3,27,28] have confirmed this hypothesis, and *S. marianum* has also shown good yield potential in these environments. In Mediterranean areas, comparatively cooler climates and higher amounts of rainfall seem to

have a beneficial effect on vegetative plant development, but at the same time, their impact on seed yield is, on average, rather small. Hence, milk thistle plants growing under cooler and more humid conditions have taller individuals, sometimes with a higher number of secondary branches and inflorescences, but at the same time, the inflorescences are smaller and bear lower amounts of seeds [3,28]. This outstanding "plasticity" of plant geometry acts as a compensatory mechanism for seed yields and can explain the large boundaries of the *S. marianum* cultivation area [64–66]. Generally speaking, in southern Europe and Asia, where winter temperatures only occasionally fall below zero, milk thistle can be grown in an overwintering cycle, whereas under cooler climates, where winter conditions are more severe, the species can be defined as an annual summer crop [21]. In such a case, the growing cycle lasts 100–130 days depending on weather conditions.

### 5.2. Cropping System

So far, scarce field experiments have been carried out to individuate the most proper place of *S. marianum* in a field rotational scheme, and further trials should address this issue. However, several studies have stated that milk thistle is not suitable for monoculture, mostly due to the root damage caused by insects' larvae [41]. In fact, this cultivation system proved to cause yield reductions of up to 40% as compared to the yields achieved in crop rotation [41]. Although the crop seems to take advantage of the residual soil fertility left after a legume, an excess of fertility (especially from N) could push the plants towards the formation of disproportionate green biomass with detrimental effects on seed production. A feasible option would be to grow milk thistle after a cereal crop, and as a break crop, it might prove useful in interrupting the repeated sowing of cereals. Furthermore, several studies have shown the high potential of this species as a predecessor of a spring-summer crop since, for instance, it is a valuable forecrop of maize, especially maize cultivated for silage [67–69].

### 5.3. Soil Management

Milk thistle is directly seeded in soil. The seedbed for milk thistle must be well-prepared [37]. Hence, it is usually plowed to a depth of 25–30 cm [24].

Sowing occurs in autumn or spring, depending on environmental conditions (rainfall and temperature) [37]. Seeds have no vernalization requirements and can germinate just after maturity [2,32].

Recommendations for sowing depth range from 1–1.5 cm to 3 cm [23], and plant emergence is highly reduced when seeds are buried at a depth of 6 cm or more [57].

A crop's optimum germination occurs between 2 and 15 °C [2], showing a decrease after this value [64]. In Poland, seed germinability was reported to vary from 65 to 75%, depending on the year [23]. Therefore, in practice, an additional 25–35% seed is required for sowing, and seeding rates used in the countries of central Europe reach about 18 kg ha$^{-1}$, a generally higher amount than in the southern regions [23,70,71]. In some cases, when the emergence of seedlings is believed to be too high, stands are thinned to the plant's desired density [72].

When the crop is intended for seed production, row spacing is usually 40–75 cm, with 20–30 cm between the plants in each row, corresponding to a plant population of between 4.5 and 12.5 plants m$^{-2}$ [33,55]. However, to limit the effects of asynchronous seed production and thus minimize seed losses at harvest, denser populations (30–50 plants m$^{-2}$) are often adopted to set a constraint on the development of secondary branches [23]. On the contrary, when cultivation aims to produce biomass, less dense stands are preferable in order to allow for the bigger development of individual plants. The influence of plant population on seed chemical composition is controversial: Omer et al. [73] reported that a narrow row spacing (25 cm) produced a reduced oil and flavonolignan content in seeds as compared with a wide row spacing (50 cm), whereas other researchers [74] did not find any related effect.

*5.4. Fertilization*

Guidelines regarding fertilization in milk thistle are variable according to the cultivation site; decisions about fertilization are strongly dependent upon environmental features (such as climate, soil moisture, and soil characteristics) as well as upon expected yields and quality traits. In rainfed cultivation, as plant nutrient use efficiency is expected to be lower, fertilizer recommendations are tuned to the average rainfall incidence, whereas when water for irrigation is available, more fertilizer is commonly used [75,76].

The species is generally considered to respond well to nitrogen fertilization.

In Poland, about 40–50 kg ha$^{-1}$ of nitrogen are normally supplied at sowing [37]; the administration of an additional 30–40 kg ha$^{-1}$ of N, in the form of ammonium nitrate, at the stage of 2–3 true leaves, proved to give the best results on seed yield through the simultaneous increase in the major yield parameters, namely the number of inflorescences per plant, the number of seeds per inflorescence, and the average weight of the seeds [77]. A similar result was obtained by Liava et al. [78], who found a significant increase in the number of inflorescences per plant and the number of seeds per inflorescence after the distribution of 125 kg ha$^{-1}$ of N.

However, the sole identification of the optimal dose of N is not sufficient for the design of a correct fertilization plan for *S. marianum*; other assessments are necessary, especially considering the spatial and temporal distribution of N fertilizers. Thus, since the needs of the crop are uneven during the growth cycle, a crucial aspect is the synchronization of the time of the release of the fertilizer with the proper times of absorption by the crop. In a study carried out by Skolnikova et al. [79] on the effect of four variants of nitrogen fertilization (lower and higher single doses, lower and higher split doses), a higher single dose of N positively influenced the number and the yield of achenes. In the same experiment, the amount of silymarin complex constituents was slightly modified according to the N fertilization treatment. This is consistent with a previous study carried out by Andrzejewska and Sadowska [80], where nitrogen fertilization did not affect the accumulation of oil and silymarin in the fruits, nor the antioxidant activity of the total phenolic content of the fruit extracts.

Phosphorus is usually supplied in an amount ranging from 30.5 to 140 kg ha$^{-1}$, according to soil conditions (especially soil pH and P availability). This element shows its best effects on the crop when supplied in association with N and K [4].

In technical literature about milk thistle, supplies of potassium have been found ranging from 50 to 150 kg ha$^{-1}$. However, although K seemed to exert a generally positive effect on several yield components, the administration of high K doses does not cause significant seed yield increases [77,80].

Similarly, positive effects resulted from the application of growth regulators in combination with soil or foliar mineral fertilizers. The treatment of milk thistle with plant-growth regulators in combination with soil or foliar mineral fertilizers increased the total production of silymarin by increasing seed yield per hectare [81].

*5.5. Irrigation*

Thanks to its resistance to drought, milk thistle is considered a typical non-irrigated crop, and in most cases, normal rainfall is usually supposed to be enough for satisfactory seed production [27,37]. Andrzejewska et al. [23] reported an average seed yield of 1230 kg ha$^{-1}$ in spring sowing during a 3-year experimental trial, with only 180 mm of rainfall during the entire crop cycle. However, in Mediterranean environments under severe drought conditions, supplementary irrigations could be advisable to enhance and stabilize seed production, especially during seed growth and filling. Crops undergoing severe water stress (50% of estimated total water requirements) exhibit lower biomass production compared to fully watered crops, and, in these conditions, stalks have much higher biomass than flowers and leaves [82]. Moreover, both water excess and deficit inhibit silymarin accumulation [23].

### 5.6. Harvest and Seed Yields

Harvest time individuation is a difficult aspect of the milk thistle cultivation technique. To minimize seed losses, harvesting should be performed before the physiological maturity of the seed, even if pharmaceutical material of good quality is obtained when achenes are fully mature [33]. A delay in harvest results in the loss of seed yield because of uneven flowering and seed ripening, and in these conditions, yield losses can reach 30–40% [83]. In one trial, seed that spontaneously fell before harvest time was high enough to allow for the establishment of a new homogeneous plant stand in the subsequent year, with yield results similar to those obtained from the properly seeded stands [28]. Otherwise, when a second cultivation year of milk thistle is unwelcome, seeds that escaped harvesting represent a source of severe weed infestation for the following crop, and big efforts are usually required for its containment below the damage threshold [29]. In this case, herbicide application on the subsequent crop is considered the most reliable control strategy [39]. Nevertheless, as seedlings and rosettes are more sensitive to selective broadleaf herbicides than older plants [29], herbicide effectiveness depends on *S. marianum*'s developmental stage, and proper timing is crucial for a successful weeding operation.

Curioni et al. [84] indicate that the best time to harvest is when 35–83% of the flower heads are dry, or at least with dry flowers and the bracts still green. Depending on growing conditions, seed yields range from 0.25 to 1.80 t ha$^{-1}$, and silymarin yields range from 10 to 40 kg ha$^{-1}$ [37,70,71,80].

### 5.7. Weed, Pest, and Disease Management

As in many other MAPs, there is also a general lack of information regarding the incidence and management of weeds, pests, and diseases in milk thistle. Indeed, the sources of information related to the diseases and pests of MAPs are mostly limited to a few cases in which their cultivation reached appreciable levels. Moreover, it is worth noting that such yield-reducing factors (i.e., weed, pests and diseases incidence) have a special relevance for MAPs in general, since their presence may not only lead to a decrease in yields, but also in the quality of production [46].

Although the plant is generally considered highly competitive, the presence of weeds may significantly reduce the harvested biomass; an experiment carried out in Greece on a 3-years basis assessed a 26% reduction in milk thistle biomass in weedy fields as compared to the weeded controls [85].

The species *Puccinia punctiformis* and *Microbotryum silybum* have been described as cryptogams that are harmful to milk thistle [86]. Powdery mildew from *Erysiphe cichoracearum* has also been seen, leaving leaves, stems, and fruits covered with whitish mycelium; likewise, lesions on leaves may be caused by *Botrytis cinerea* [87].

Among insects potentially harmful to the crop, the weevil beetles *Larinus latus* [88] and *Tanymecus palliatus* [87] have been observed. Aphids have occasionally been detected (Carrubba, pers. observ.); their infestations mainly involve leaves and young buds, but flowers and all tender tissue may be attacked. Infested plants exhibit a variety of symptoms, such as spotted or yellowing leaves, curled leaves, wilting, decreasing growth rates, low yields, and finally, death [87,88].

## 6. Qualitative Characteristics of Seeds

At physiological maturity, the fruit ("seed") of *S. marianum* contains about 7% water, 20–30% lipids, 20–30% proteins, 0.038% tocopherols, 0.63% sterol and other compounds such as 3-deoxyflavonolignans mucilage [9,89].

According to Malekzadeh et al. [90], the most abundant oil constituents are oleic (36.7%) and linoleic acid (39.7%), while other less abundant fatty acids are palmitic (10.2%), stearic (6.9%), arachidic (3.6%), and behenic (2.5%), along with trace amounts of linolenic and eicosenoic acids. High variability among genotypes was observed [91,92] in the available content of oleic (between 20.9 and 38.2%) and linoleic acid (between 34.0 and 60.3%), whereas substantial intraspecific homogeneity was assessed in the values of the

other fatty acids present in the oil. Environmental conditions can also affect the content and the composition of the milk thistle fatty acid profile. For instance, water stress conditions determine an increase in unsaturation level, with an enhancement of the content of linoleic acid at the expense of oleic acid [90]. Additionally, other researchers [93] showed that soil fertilization had a positive impact not only on fruit yield and silymarin content, but also on the proportion of unsaturated fatty acids.

The oil also includes significant percentages of phospholipids, phytosterols, and vitamin E, which enhance its nutritional value [94–96].

### 6.1. Silymarin

Silymarin, the bioactive compound responsible for the phytotherapeutic and pharmacological properties of milk thistle, was first isolated by Wagner et al. [97]. This compound is rather a mixture of several flavonolignans, endowed with different levels of biological activity [98], among which the main constituents are: silycristin, silydianin, silybin (diastereoisomers A and B), and isosilybin (A and B) [99] (Figure 3). Silybin (accounting for 50–70% of the mixture) is the major silymarin component, followed by silychristin, silydianin, and isosilybin [100,101].

Silybin (synonymous with silibinin), the most bioactive ingredient of the milk thistle extract [36,50], was first isolated and established by Pelter and Hänsel [102]. It is considered safe, with no severe side effects except for a minor gastrointestinal disturbance, and due to its very low general toxicity, the US Food and Drug Administration (FDA) approved its use as a phytomedicine to treat liver diseases [103].

High intraspecific variability in the relative amounts of silymarin constituents has been noted by many authors [104–107]. Martinelli et al. [92] showed that the amount of the individual flavonolignan in the silymarin mixture was the most variable trait among 26 accessions collected from different sites. Research on the phytochemical variability of milk thistle first allowed for the identification of two high-silybin and high-silydianin chemotypes [108]. In a later work [109], the same research group listed three chemotypes (termed A, B, and C) within Italian wild populations, with the chemotype C being the result of the hybridization between A and B.

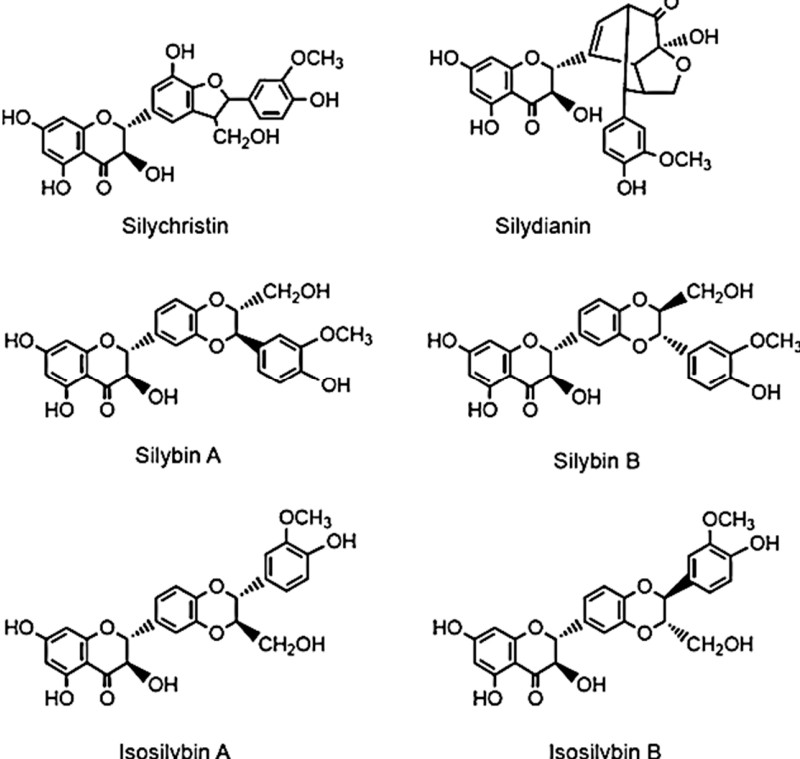

**Figure 3.** Major constituents of silymarin. (Adapted from [110]).

### 6.2. Biosynthesis and Accumulation of Silymarin

Considerable research has been devoted to studying the biosynthesis process of the flavonolignans of milk thistle. In the first step, they are synthesized through the phenyl-propanoid pathway by the oxidative radicalization of two precursors, namely a flavonoid (mostly taxifolin) and coniferyl alcohol; the final step consists of the coupling of these two radicals [107,111,112]. The biosynthesis of taxifolin occurs in the flower, then it is transferred to the pericarp where the synthesis of silymarin's constituents takes place [113]. The main silymarin storage site is the seed integument, whereas only trace amounts are present in the pericarp layer [114]. However, the amount and the accumulation of silymarin in the seed integument of milk thistle are significantly variable and deeply affected by environmental conditions; research has shown that drought stress conditions can enhance the biosynthesis of this active substance in plant tissues [115], and that its accumulation in the fruits is related to the growth phase and the lignification of the seed coat [33,105].

According to the European Pharmacopoeia and the United States National Formu-lary, mature fruits of *S. marianum* must yield a minimum of 1.5–2% silymarin on the dry matter [105]. However, there is evidence that silymarin content in seeds can be strongly variable according to genotype and growth conditions, since values ranging from 0.62 to 2.25% [108], from 2 to 4% [107], and sometimes amounts as high as 4.2% [104] and 6% [107] have been reported.

Due to the high pharmaceutical relevance of silymarin and the increasing demand of pharmaceutical companies for steady and homogeneous supplies of silymarin compounds, recent efforts have focused on different approaches to maximize and stabilize its production, including effective extraction methods and biotechnological production [116].

### 6.3. Extraction Methods

Due to the high lipid content (20–30%) in milk thistle seeds, extracting silymarin using a single-step procedure is quite a difficult task. Hence, the European Pharmacopoeia suggests a two-step solvent extraction (SE): first, the defatting of the seeds for 6 h in n-hexane, followed by silymarin extraction using methanol for five additional hours [117].

Nevertheless, this extraction procedure has various drawbacks, mostly related to the exceedingly long duration of the process and the high toxicity of hexane. Therefore, alternative methods for silymarin extraction have been explored. A significant shortening of the process can be achieved by using the technique of pressurized liquid extraction (PLE) [117], which permits the avoidance of the preliminary defatting, otherwise required by the traditional method. Additionally, new technologies such as microwave-assisted, ultrasound-assisted, and enzyme-assisted extraction have been studied to increase the extraction yield of silymarin [118].

### 6.4. Biotechnological Production

The strong interest in silymarin as a pharmaceutical item has pushed for research into its biotech production. This procedure could be an effective alternative method to make silymarin production continuous and stable over time, while also overcoming the issues inherent to conventional open-field production [119] as well as the high intraspe-cific variability of silymarin seed content [104,107,120]. Tissue and cell cultures as well as in vitro regeneration techniques, with or without the addition of elicitors to the growth medium [121,122], were applied with this objective. Studies using in vitro cultures of *S. marianum* began in the late 1970s [123] and went on, with contrasting results, through-out the last decades of the twentieth century, up to the present [110,124–128]. In most cases, in vitro techniques allowed for low silymarin yields to be compared with field production [110,124–128]. Better results were otherwise obtained through a bioreactor sys-tem [129] consisting of the mass cultivation of *S. marianum* hairy roots, for the production of silymarin on a large scale. However, this technique is still far from industrial application, and the specialized cultivation of the starting plant material is so far the most reliable way to produce silymarin. For future development, further efforts must be focused on

optimizing the physical and chemical conditions for in vitro cultures of milk thistle in bioreactor systems [107].

## 7. Utilizations

Medical applications of milk thistle have been thoroughly demonstrated by many experiments for more than 50 years [5–8,12,14–16,19,20,116,130–135], and interest in silymarin has increased in the fields of medical, pharmaceutical, and veterinary sciences due to its wide use in various therapies and medical applications [18,101,136,137]. However, the yield potential and the chemical composition of milk thistle fruits and vegetative biomass have suggested a variety of additional uses, ranging from livestock feeding to human consumption and industrial exploitation, and involving several plant parts, derivatives, and extracts (Tables 1–3).

**Table 1.** Major observations on the actual and potential uses of milk thistle (*Silybum marianum* (L.) Gaertn.) for feeding purposes in different livestock species.

| Livestock Species | *S. Marianum* as Fodder or Feed Additive | Present and/or Potential Uses | References |
|---|---|---|---|
| Sheep | Plant (green forage) | Dry matter intake, rumen fermentation parameters, and blood metabolites were positively affected by diets containing *S. marianum*, proving its suitability to the feeding of small ruminants. | [138] |
| | Seed (oil) | Oil from *S. marianum* proved to influence rumen fermentation, improving protozoa counts in sheep rumen. | [139] |
| | | Ruminant diets with oils derived from *S. marianum* plants enhanced desirable milk fatty acid (FA) profile and milk yield. | [140] |
| | Seed (silymarin extract) | A higher milk production throughout lactation was detected, with no evidence of toxic effects in ewes. | [141] |
| Dairy cow | Plant (green forage) | Forage from *S. marianum* appeared to be worse than green forage from barley. | [142,143] |
| | Plant (silage) | *S. marianum* could be used as a source of flavonolignans, beneficial for health, milk yield, and milk quality in dairy cattle. | [143] |
| | Seed (whole/ground) | Supplement from milk thistle endosperm considerably increased the concentration of monounsaturated fatty acids in milk fat. | [144] |
| | Seed (flour) | *S. marianum* seed flour proved to be a suitable feed additive for young ruminants. | [145] |
| | | Fermented solid wastes from milk thistle seed flour showed good suitability for the improvement of feed quality. | [146] |
| | Seed (oil) | Oil from *S. marianum* showed to influence rumen fermentation, improving protozoa counts in cows' rumen. | [139] |
| | Seed (silymarin extract) | Earlier peak of milk production and higher milk production throughout lactation were detected, with no evidence of toxic effects in dairy cows. | [147] |
| | | The hepatoprotective effect on dairy cows was detected. | [147] |
| | | Silymarin supplementation to feed rations speeded up the transition to the metabolic adaptation of dairy cows at the beginning of lactation, with no negative side effects. | [148] |

**Table 1.** *Cont.*

| Livestock Species | *S. Marianum* as Fodder or Feed Additive | Present and/or Potential Uses | References |
|---|---|---|---|
| Buffalo | Plant (dried and powdered) | Milk thistle could be used in up to 20% of the buffalo's diet without any negative effect on digestion and fermentation characteristics of whole rumen microorganisms and bacteria. | [149] |
| Broiler chicken | Seed (whole/ground) | The introduction of *S. marianum* in starter and grower rations allowed for the highest body weight at the lowest feed conversion per body weight gain unit without influencing muscle mass and fattening grade, at the same time improving the meat's nutritional value. | [150] |
| | | *S. marianum* seeds showed good hepatoprotective efficacy in counteracting the toxic effects of food contamination with aflatoxin B1. | [151–154] |
| | Seed (extract) | The use of *S. marianum* fruit extract in broiler chicken nutrition enhanced the performance and meat quality. | [155] |
| | Seed (silymarin—phospholipid complex) | Protection against the negative effects of aflatoxin B1. | [156] |
| Goat | Plant (green forage) | Evidence of the degradability/digestibility of *S. marianum* seeds and the diffusion of the species in local pasture areas through natural grazing. | [157,158] |
| | | Goats consumed up to 99% of the cut *S. marianum* foliage, also underlining the potential to include the grazing of goats in an integrated weed management strategy for the crop. | [159] |
| | | Goats ingest the *S. marianum* pasture readily. | [160] |
| | | Goats readily included the capitula of *S. marianum* in their diet, especially before maturity, determining the quick reduction of fodder availability and the potential seed production. | [161] |
| | Seed (oil) | Oil from *S. marianum* was observed to influence rumen fermentation, improving bacteria counts in the goats' rumen. | [139] |
| Pig | Seed (whole/ground) | *S. marianum* seeds appeared to be a useful feed admixture for fattened pigs to improve meat quality, oxidative stability, weight gains, feed utilization, polyunsaturated fatty acid content in tissues, and water holding. | [162] |
| Rabbit | Seed (whole/ground) | Flavonolignans and taxifolin showed a mild effect on the growth performance of rabbits, and the diet containing the highest amount of *S. marianum* constituents was able to attenuate the morbidity and mortality of broiler rabbits. | [163,164] |
| | | Dietary supplementation with *S. marianum* changed the sensory characteristics of rabbit loin. | [164] |
| Common carp | Seed (silymarin extract) | *S. marianum* appeared to enhance some nonspecific immune responses. | [165] |
| Rainbow trout | Seed (silymarin extract) | Reduction of plasmatic glucose and cholesterol, with side effects on blood biochemical and clinical parameters only when doses were high, were detected. | [166] |

**Table 2.** Major observations on the actual and potential uses of milk thistle (*Silybum marianum* (L.) Gaertn.) for food purposes.

| Plant Parts, Products, and By-Products | Present and/or Potential Uses | References |
|---|---|---|
| Leaves, young stems, and sprouts | Young fleshy stems and sprouts are traditional food items in several Mediterranean countries. | [167] |
| | *S. marianum* is a good candidate source of healthy edible sprouts. | [64] |
| | Young heads and stems are traditionally eaten in Sardinia. | [168] |
| | In Spain, traditional use as a fresh vegetable in salads or consumed boiled and fried has been reported. | [169] |
| Whole seed (unprocessed) | High amounts in healthy substances such as fibers, and Ca and K levels, even higher than many conventional vegetables, was detected. | [170] |
| Seed (processed/fermented) | Effective antioxidant and radical-scavenging activity of silymarin from seeds suggests its utilization for minimizing or preventing lipid oxidation in food products. | [171] |
| | Fermented *S. marianum* seeds proved to be a suitable additive for the natural flavoring of baked goods. | [172,173] |
| | *S. marianum* protein had an excellent balance of all essential amino acids, with potential application as a functional food ingredient. | [174] |
| Seed (flour) | *S. marianum* seed flour was successfully tried for the enrichment of biscuits. | [175] |
| | *S. marianum* seed flour could be added to functional foods with potential health benefits due to its phytochemical composition and gut microbiota-modulating, free radical-scavenging, anti-inflammatory, and anti-proliferative properties. | [176] |
| | Partial wheat flour replacement with *S. marianum* seed flour was suggested for the production of healthy biscuits, providing for a new item in functional bakery products on the market. | [177] |
| Seed (oil) | *S. marianum* seed oil was suggested as a substitute for highly unsaturated fatty oils for human consumption. | [178] |
| | *S. marianum* seed oil can be used in food and industrial foods as well as cottonseed, sunflower, and soybean oils. | [31] |
| Seed (silymarin extract) | The introduction of pure silymarin in food could hamper the formation of toxic oxidation products, helping to maintain nutritional quality and prolonging the shelf life of food. | [175] |
| | When used as a food stabilizer, silymarin exhibited a protective effect against arsenic-induced cytotoxicity. | [171] |

**Table 3.** Major observations on the actual and potential uses of milk thistle (*Silybum marianum* (L.) Gaertn.) for industrial non-food/non-medicinal purposes.

| Plant Parts, Products, and By-Products | Present and/or Potential Uses | References |
|---|---|---|
| Whole plant (crop/unprocessed) | Good biomass productivity suggested utilization as a bioenergy source even in Mediterranean environments. | [179,180] |
| | *S. marianum* lignocellulosic biomass was tested for biogas production with positive results. | [181] |
| | Whole plants or residues after seed harvest were submitted to anaerobic fermentation, indicating *S. marianum* as a suitable raw material for biogas production. | [21] |
| | The plant's ability to grow in soils with a wide range of heavy metal contamination, where it behaves as a metal excluder or a tolerant plant, suggested its utilization for phytoremediation purposes. | [182–187] |
| Various plant parts (extracts) | Byproducts of the pharmaceutical processing of *S. marianum* seeds were reported to have insecticidal properties against the green peach aphid and the greenhouse whitefly. | [188] |
| | Bioherbicides and biodegradable pesticides, useful in organic farming, were obtained from *S. marianum* seed, flower, stem, leaf, and root extracts. | [189] |
| Seed (flour) | A bacteriostatic effect was detected, allowing *S. marianum* to be adopted in phytosomal formulations used as functional cosmetics. | [190,191] |
| Seed (oil) | Oil extracted from *S. marianum* seeds proved suitable for biodiesel production. | [192,193] |
| | *S. marianum* seed oil and other mixed ingredients proved effective in the treatment of facial wrinkles, improving skin elasticity, dermal density, and tone. | [194] |
| | Cold-pressed *S. marianum* seed oil could be applied to the skin as cosmetics, with no irritating effect and rare allergic reactions. | [195] |
| Seed (silymarin extract) | Antioxidant and UV-protection activity of silybin allowed for the use of *S. marianum* derivatives in the preparation of cosmeceuticals for skin protection. | [146,196] |
| | The excellent toxicological and bioactivity profile of silymarin in a glycerol suspension as well as its anti-inflammatory activity, indicated that *S. marianum* glycerol extracts can be used in the preparation of high-value anti-aging products in cosmetics. | [197] |
| | Ultrasound-assisted extraction (UAE) of silymarin from mature fruits of *S. marianum* allowed for the green, efficient extraction of flavonolignans with potent antioxidant and anti-aging activity, to be employed in the development of cosmetic products. | [198] |
| | Topical treatments with a w/o emulsion from *S. marianum* extracts proved to possess skin whitening properties, promoting a significant decrease in skin melanin level. | [199] |

### 7.1. Livestock Feeding

The use of *S. marianum* as a livestock feeding source has been studied by numerous authors (Table 1). In many cases, the plant has shown promising results as a source of bioactive compounds that is able to reduce metabolic and oxidative stress in animals, also representing a useful tool for the enhancement of the productive and qualitative performances of livestock, including milk yield and meat quality. The long history of milk thistle as a medicinal plant can explain its long use as a food supplement in ruminants' diet, deemed capable of reducing the damage from the administration of aflatoxin-contaminated fodder [136,141,156]. The use of milk thistle within the feeding ration would represent a

supplement of flavonolignans, responsible for the increase in milk production and quality as well as beneficial for animal health and overall productivity [142,143]. Several experiments carried out in Sicily [141] highlighted the advantages derived from the introduction of milk thistle into the diet of *Comisana* ewes. The research highlighted the positive effect of silymarin extract on the production and the amount of somatic cells in sheep milk from the intermediate stage to the end of lactation. Otherwise, in livestock fed with unprocessed milk thistle seed, no improvements in milk quality were observed, probably due to the lower level of silybin intake by animals when milk thistle was administered in this form [141].

However, although promising prospects for the use of livestock feeding have emerged, studies on ruminant nutrition have highlighted some drawbacks related to the massive use of *S. marianum*. Piłat et al. [142] pointed out that digestibility, energy value, and intake by animals of green forage from *S. marianum* were lower compared to other types of green forage (e.g., barley). Krížová et al. [200] showed that the digestibility of silybin in dairy cows was 40.0–45.5%, and the degradation of all flavonolignans ranged from 23.3 to 35.2%. Another crucial aspect was reported by other researchers, who evidenced the potential antinutritional characteristics of milk thistle due to the strong correlation between soil and leaf nitrate content; given the well-known potential toxicity of nitrate, this might represent a strong limitation to the widespread use of this plant as a feeding resource [29–31,201,202].

### 7.2. Food Uses

*S. marianum* is recognized as a safe plant [203], and its young fleshy stems and sprouts were, and still are, traditionally eaten in several countries in the Middle East and North Africa that border the Mediterranean basin [167]. Different parts are consumed in Sardinia and Spain; for instance, young heads and stems [168]. In Spain, milk thistle has been traditionally used as a fresh vegetable in salads, or eaten boiled and fried [169]. These observations suggest the exploitation of its fresh leaves, without spines, as an interesting commercial opportunity [169]. The high amount of healthy components such as fibers, as well as the relevant Ca and K levels, even higher than in many conventional vegetables, represent important aspects supporting this opportunity [170].

The fruits of *S. marianum* have been recommended for food use as well. They can be used either for oil extraction or as flour, in a mixture with wheat flour, in the preparation of various bakery products [175]. Furthermore, due to their high amount of healthy unsaturated fatty acids, the presence of flavonolignans, and the high proportion of fibers, milk thistle derivatives have been suggested for the enrichment of food products [21]. Many health benefits have been claimed by such products, said to retain the antioxidant and radical scavenging properties of pure silymarin [171]. Research has demonstrated that the addition of milk thistle extracts and derivatives as well as pure silymarin to food can make the oxidative process slower, helping to maintain nutritional quality and prolonging the shelf life of food [204]. Finally, in addition to its beneficial role as a food stabilizer, the introduction of milk thistle extracts into the human diet has been shown to produce a protective effect against arsenic-induced cytotoxicity, which is a serious issue, particularly in heavily polluted areas [204].

However, as animal feed, and also in human food, nitrate accumulation in the edible parts of *S. marianum* can raise quite a few concerns [205]. For this reason, a proper agronomic technique must be developed to control the nitrate content of *S. marianum* vegetative biomass, allowing to extend prospects for its use in the human diet.

### 7.3. Industrial and Other Non-Medical Applications

Thanks to the numerous properties found among its bioactive components, milk thistle also proved to be useful for several industrial non-medical applications (Table 3).

7.3.1. Cosmetic Industry and Bioenergy Production

The antioxidant activity of silybin [196] as well as the UV-protection activity of silymarin [146] allow *S. marianum* derivatives to be successfully utilized in the preparation of

cosmeceutical products for skin protection [196]. Many of these products are already on sale for sun protection as well as for body and skincare, and the cosmetic industry also takes advantage of the bacteriostatic effect detected in milk thistle seed flour [190]. In some cases, the bioavailability of the active molecules has been improved with innovative techniques, for the controlled release of the active principles such as that registered by Indena S.p.A. (Milan, Italy) as PHYTOSOME® [191].

Bioenergy production is also another possible use of *S. marianum*. Biomass production for energy purposes has especially been studied in Mediterranean environments, where the species demonstrated to be capable of achieving satisfactory biomass yields even in low or moderate input conditions. In two trials carried out in Sardinia [179,180], the crop averaged 16–20 t ha$^{-1}$ of total dry biomass, with significantly higher productivity than *C. cardunculus* var. *scolymus* and *C. cardunculus* var. *altilis*, which had been used as control crops in the two experiments, respectively. In a 3-year experiment, milk thistle produced a total of 20.89 t ha$^{-1}$ dry biomass, with a gross calorific value ranging from 15.5 to 16.1 MJ kg$^{-1}$ [85]. Moreover, the output to input energy ratio for *S. marianum* yield was more favorable than in other crops, typically farmed for energy purposes [179].

Milk thistle lignocellulosic biomass was tested, with positive results, as a raw material for biogas production by submitting the whole plants or their residues to anaerobic fermentation after seed harvest. [21,181].

According to recent studies [192,193], the oil extracted from *S. marianum* seeds also makes this plant very attractive for bioenergy production even though the high level of free fatty acids (FFA) in the oil requires a two-step trans-esterification. The partitioned process allows for the reduction of the FFA content of *S. marianum* oil to a safe limit, thus enabling its conversion into biodiesel using an alkaline catalyst [192].

### 7.3.2. Biocidal Activity

The use of milk thistle extracts for manufacturing biodegradable pesticides and herbicides, suitable for organic farming, was suggested by several specific experiments. The ethanol extracts, obtained as byproducts from the pharmaceutical processing of *S. marianum* seeds, were reported to have insecticidal properties against the green peach aphid and the greenhouse whitefly [188]. Likewise, the production of bioherbicides from seed, flower, stem, leaf, and root extracts was studied [189]. Experiments were carried out to assess the effects of water extracts from whole *S. marianum* plants [85] on the germination of seeds of rigid ryegrass (*Lolium rigidum* Gaudin) and little seed canarygrass (*Phalaris minor* Retz.); milk thistle caused 81.7% and 47.4% reductions in rigid ryegrass germination and root length, respectively, acting in a dose-dependent fashion.

However, there is still uncertainty about the effectiveness of these extracts, as well as about the actual allelopathic effect of *S. marianum* on the germination and growth of other plant species [64], which leaves room for further experimentation aimed at filling the gap in knowledge within this field.

### 7.3.3. Phytoremediation

Several studies have also shown a further alternative use of the crop in agronomic techniques for phytoremediation. Indeed, *S. marianum* often functions as a metal excluder or tolerant plant, being able to grow and develop in soils contaminated with a wide range of heavy metals such as cadmium, chromium, lead, copper, manganese, zinc [182–184], and radioactive cesium [185]. Hence, the crop was proposed for biomass production in contaminated sites, allowing for concomitant bioenergy production and phytoremediation [186,187].

## 8. Conclusions

Although silymarin surely represents the major point of interest in milk thistle so far, the latest research has shown that this crop could also find several industrial uses, which could dramatically enhance its economic importance. Food and feed use, bioenergy

production, cosmetic and cosmeceutical applications, and phytoremediation are only a few out of many novel opportunities offered by the medium-large scale cultivation of milk thistle. The environmental suitability of the species is wide, as was demonstrated by the good yield levels obtained in a range of different soil and climate conditions. Hence, the opportunity arises to cultivate it even in marginal (or environmentally constrained) conditions such as those typical of many semi-arid Mediterranean environments, where the search for suitable solutions for economical and agronomical diversification of agroecosystems is a serious issue. Indeed, the biological and agronomical traits of this plant (late-spring flowering, very low fertilizer requirements, drought resistance, and good suitability to poor soils) make it a reliable alternative in low-input agriculture, one that is able to offer good production in sustainable agricultural systems. However, additional research efforts have to be performed with this purpose: the species is still characterized by morphological and physiological traits belonging to non-domesticated plants, which bring many difficulties in the application of some of the most common agronomic practices. Among these, the plant's spiny habits, the asynchronous flowering/ripening stages, and the often-unpredictable reactivity to technical inputs such as fertilization and irrigation are all major issues that further genetic and agronomic research must address. There is an urgent need for reliable field data devoted to the definition of suitable cropping protocols, addressed to the different goals of production (biomass or seed), in order to prevent the slowdown in the spread of such a promising crop. Likewise, research on some novel industrial applications of the species is lacking, such as those involving the exploitation of its biocidal/herbicidal potential. This should involve public investments, supporting both research activities and farmers.

**Author Contributions:** Conceptualization, A.C., M.S. and G.D.M.; methodology, A.C., M.S. and G.D.M.; investigation, R.M. and L.D.; resources, G.D.M.; writing—original draft preparation, A.C., R.M. and L.D.; writing—review and editing, A.C., R.M., L.D., M.S. and G.D.M.; visualization, A.C., R.M., L.D., M.S. and G.D.M.; supervision, A.C., M.S. and G.D.M.; project administration, G.D.M.; funding acquisition, G.D.M. All authors have read and agreed to the published version of the manuscript.

**Funding:** This research was funded by the regional project "COSA—Conservazione e caratterizzazione di Accessioni Siciliane di Specie Agrarie erbacee", Misura 10, Sottomisura 10.2, Operazione 10.2a, PSR Sicilia 2014–2020.

**Institutional Review Board Statement:** Not applicable.

**Informed Consent Statement:** Not applicable.

**Data Availability Statement:** The data presented in this study are available on request from the corresponding authors.

**Conflicts of Interest:** The authors declare no conflict of interest. The funders had no role in the design of the study; in the collection, analyses, or interpretation of data; in the writing of the manuscript, or in the decision to publish the results.

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
