# Peer review of "Milk Thistle (Silybum Marianum L.) as a Novel Multipurpose Crop for Agriculture in Marginal Environments: A Review"

_agronomy, doi:10.3390/agronomy12030729_

Round 1

Reviewer 1 Report

The manuscript is clear, well written, and easy to understand for readers. Information presented in this review are well systematized. Authors present in a succinct way the state of art regarding the origin of S. marianum, the management techniques regarding fertilization needs weed, pest and disease management,  qualitative characteristics of seed ( in connection with extraction methods,  and biotechnological productions), their utilization in livestock, human food, and industrial applications.    This plant or some parts from it can be used as a raw material in the cosmetic industry, biomass can be processed to obtain bioenergy and biocidal products.  Current knowledge and limitations of it are presented clearly and concisely in this manuscript. 
For scientists this manuscript is interesting and with certitude will open new ways in applicated research;

In my opinion, minor additions must be made in the manuscript before publication, respectively: 

a) At figure 1 and Figure 2 ( A, B) authors must write if these are original or in the case in which these were taken from another source, they must supply the permission to reuse it.

b) At row 67, authors must write in the brackets what does it mean ''m a. b.l.''

c) At row 74, the authors must write in the brackets what does it mean ''A.D''

d) At row 310,  the authors must write in the brackets what does it mean''MAPs'

e) At  row 374 the word ''much research'' must be replaced with the word ''more research''

f) At row 472 the row ''facing'' must be replaced with another word.

Reviewer 2 Report

Dear authors,

  1. I have read the article carefully. But considering the scope of using underutilized crops for agricultural sustainability as well as versatile use the topic can be considered.
  2. A copy-pest article.

            Original source file: Karkanis et al. (2011) doi:10.1016/j.indcrop.2011.03.027

The concept, presentation style and information are based on the above source. A shadow of the mentioned article is very prominent inclusive of plagiarism.

  1. Expand the geographical location from the Mediterranean region to other regions/ continents where there is possibility/ suitability to grow the crop. It will further make the article different from the earlier published article [doi:10.1016/j.indcrop.2011.03.027]
  2. Any data related to the economic feasibility? If not, then why the same can be considered by replacing the existing competitive crop?
  3. The article in its present form may be rejected because of the following points; however, considering the future scope of the crop, the authors may be given a chance to RE-CAST the same in the following line with latest and relevant information on the subject.
  4. The authors should mention the novelty of the article.
  5. Further, the authors are requested to consider the following specific issues.

Title:

  1. The title should be changed as nowhere sustainable agriculture has been addressed and described.
  2. Why in semi-arid environment? It has wider distribution.

Alternative title may be:" Milk thistle, a versatile and underexploited crop needs wider extension" Or something else can be thought. [by broadening the area of adaptation, scope etc]

Abstract:

Why Milk thistle can be grown only on Mediterranean region? Any specific reason(s)? Why not in other areas/ continents?

Management Techniques:

  1. In management techniques, cropping system has not been mentioned.
  2. Whether it will replace existing crop(s) or grown in fallows?
  3. Other than Mediterranean region, in which cropping system, the crop will be suitable?

# Attachment: Plagiarism report

All the best!

Reviewer 3 Report

Reviewer comments

Agronomy (ISSN 2073-4395)

Manuscript ID: agronomy-1623124

Manuscript Title:Milk thistle, an alternative crop for sustainable agricultural systems in semi-arid environments. A review"

In this study, the authors wrote a review article about “Milk thistle as an alternative crop for sustainable agricultural systems under semi-arid environments” The topic is nice and fits well with the scope of agronomy -MDPI, and this point is interesting for the scientific community. The tables and figures (Photos) are prepared in good quality and well structured. However, the text needs a major revision before publication.

The comments:

Abstract:

  • The abstract part is well written, the authors mentioned a comprehensive introduction to the plant and its uses in general, and the status of the plant from the field point of view, but this abstract lacks some information that may be important, so can the authors give examples from the challenges that facing plant growth, spread, and availability?

Keywords: Pls rewrite it in a simple form avoiding repetition, especially regarding the words that are already present in the title, if possible.

Introduction:

  • Line 32: “Milk thistle (Silybum marianum (L.) Gaertn.) is an annual or a biennial herb [1,2]. Please explain when the plant is seasonal or biennial according to previous literature in this concern.
  • Lines 43-45:” In recent years, research into the use of silymarin expanded considerably, also embracing the possibility to cure other illnesses and ailments. In addition to its hepato-protective effect [15]. The ref. No. 15 was 2014 and this is not recent as mentioned by authors.
  • Lines 69-70: “Milk thistle has long been known as a useful species. Archeological surveys have shown that this type of thistle was probably already used in the Neolithic Era in the Mediterranean environment [31]. Pls shorten.
  • Lines 78-81: “In natural conditions, milk thistle is commonly referred to as a ruderal species, as dense stands can be found in fertile, highly disturbed environments, such as roadsides, pastures, or sheep camps, especially when endowed with high soil nitrogen levels [35]. Furthermore, its spreading in cultivated fields can be facilitated by special plant’s traits, such as the remarkable seeds production and viability, and the easy wind dispersal, due to the long feathery seed pappus”. Pls rephrase and shorten.
  • Lines 87-93: “In Pakistan, wheat yield losses caused by S. marianum infestation ranged from 7% to 37% [39]. In pastures as well, it is considered a dangerous invasive species, since its large rosette (up to 1 m in diameter) has the potential to displace most other pasture species, and its spiny thistles can hinder the movement and grazing activity of livestock. Additionally, due to the plant’s ability to accumulate nitrogen, some cases of livestock intoxication after milk thistle ingestion have been reported, especially when the ingested plant was in the early wilting stage [35,37,39]”. This part is not suitable in this place; it must be moved to the appropriate place, which is after mentioning the benefits of the plant. It is also harmful, especially on the plants growing next to it, which must limit its spread or harvest it to benefit from it medically as mentioned previously.
  • Lines 95-96: “The genus Silybum belongs to the family Asteraceae and groups 2 species: marianum (L.) Gaertn., and S. eburneum Coss. & Durieu [41,42]”.
  • The meaning of this sentence is repeated in more than one place, including the abstract part. Therefore, it must be deleted while avoiding repetition in the entire manuscript.
  • Lines 104-106: “Despite the increasing interest in this species as a multipurpose crop, and its actual economic importance in the herbal market, the plant has not been subjected to a thorough breeding activity as far”. Pls add relevant ref.
  • All images in figures are taken from the Internet without mentioning the source of the images or the site from which the images were taken. This is not good, and it was very easy for authors to put pictures of plants from their photography, especially since plants are available and easy to obtain and photograph from nature.
  • Lines 141-142: 4. Description, 4.1. Morphology, pls shorten this whole part
  • The current state of this review article needs some more attention.
  • The usage of abbreviation should be used after the full term. Please be consistent with the usage of all abbreviations. Pls revise the abbreviations in the whole parts of this review article.
  • Lines 193-196: The fourth stage includes seed growth and filling. Fruits are white/ivory colored at 193first (BBCH stage: 7.1), and assume a darker hue throughout ripening (BBCH stages from.5 to 7.9). The seeds are ripe in July. During fruit ripening, also plant senescence begins. Usually, once fruits on the side branches are ripened the entire plant appears completely 196dead and dry [62]. Please shorten these sentences”.
  • Lines 216-218: This outstanding “plasticity” of plant geometry acts as a compensatory mechanism for seed yields and can explain the large boundaries of marianum cultivation area. Pls add ref.
  • Line 249: 5.3. Fertilization. Pls be specific in this regard.
  • Lines 288-291: The individuation of the most appropriate harvest time is a difficult element of milk thistle cultivation technique. In order to minimize seed losses, harvesting should be done before the physiological maturity of the seed, even if pharmaceutical material of good quality is produced when achenes are fully mature [28]. Please shorten these sentences.
  • Lines 309-319: 5.6. Weed, pest and disease management “As in many other MAPs, also in milk thistle there is a general lack of information regarding the incidence and management of weeds, pests, and diseases. This issue is mostly due to the limited cropping area of such crops. In fact, the sources of information related to the diseases and pests of MAPs are mostly limited to areas in which their cultivation reached appreciable levels. Nowadays, in view of the increasing interest in MAPs cultivation throughout the world, such an issue is gaining deeper attention, and the pests and diseases control techniques have taken great relevance inside the recommended growing protocols for these species. It is worth noticing, moreover, that such yield-reducing factors take a special relevance for MAPs, since their presence not only may lead to a decrease in yields, but also in the quality of production [70]”. Please shorten these sentences.
  • Lines 410-416: 6.4. Biotechnological production “The strong interest in silymarin as a pharmaceutical item has pushed towards research into the production of this compound through a biotechnological approach. This procedure, in fact, could be an effective alternative method to make silymarin production continuous and stable over time, also overcoming the issues inherent to conventional open-field production, such as the uneasy agricultural management, the strong reliance of productions on erratic seasonal trends [113]”. Please shorten these sentences.
  • General comment; Pls, shorten the long sentences and add Ref. as soon as possible and avoid repetition.
  • Conclusions; Do not repeat the above sentences in the conclusion part. In conclusion, you should write a summary of your work in short sentences so that I, as a reader of the reference article, can understand what the article ended up being.
  • There is another important part that must be added at the end of this review article, which is the current situation of using this plant and its production in safe form and adding the vision or future expectations of using this plant or its extracts in order to increase the productivity as well as from an economic point of view and in terms of its impact on protecting the environment, ...etc.
  • References; pls revise the Ref. carefully.

General comments:

  • The manuscript contains some typo errors; please revise it very carefully. A careful revision of the English grammar is required.

Round 2

Reviewer 2 Report

Dear authors,

The modified version is better addressing the points I raised. I appreciate your efforts. Please look into some suggestions made in the manuscript and consider the following points.

  1. Plagiarism is still high and not ok for a review article.
  2. In Management Techniques, a paragraph/ subheading should be included as "cropping system" in which it can be mentioned that in which system the crop is to be considered. What are the competitive crops? Whether it has the potential to replace the fallowness? etc. In this regard, please consult to any agronomist.
  3. Once you look into the references arrangement and it should be as per the journal style.
  4. Thanks & regards.

Author Response

Response to Reviewer 2 Comments

Point 1: Plagiarism is still high and not ok for a review article.

Response 1: Again, we are entirely sure that no plagiarism occurred. After several small changes in the text, a second anti-plagiarism report gave a total matching of 21%, lower than the previous result. We made other small changes and probably a third report would give even a lower outcome, but since all individual matches are lower than 1%, and most of them are boilerplate text (the name of the species, the name “silymarin”, even the Publisher’s note), in our opinion this issue can be considered as solved.  

Point 2: In Management Techniques, a paragraph/ subheading should be included as "cropping system" in which it can be mentioned that in which system the crop is to be considered. What are the competitive crops? Whether it has the potential to replace the fallowness? etc. In this regard, please consult to any agronomist.

Response 2: Thank you for this suggestion. We have included a new paragraph that deals with the “cropping system” (lines 235-248). We all are agronomists, and we are well aware of the importance of the issues you raised; however, we believe that a reliable answer to these questions can be obtained only after that all general supporting points have been defined. Hence, further suggestions for a potential role of this crop within the common agroecosystems have been provided in the “conclusions” section (see in particular lines 600 to 619).

Point 3: Once you look into the references arrangement and it should be as per the journal style.

Response 3: Thank you for this suggestion. A complete check was carried out and all the issues concerning the references’ arrangement have been fixed.

Reviewer 3 Report

Reviewer comments

Agronomy (ISSN 2073-4395)

Manuscript ID: agronomy-1623124

Manuscript Title:Milk thistle, an alternative crop for sustainable agricultural systems in semi-arid environments. A review"

After revising the entire review article and matching the authors' response to my previous comments on it, I believe that although the authors did not respond to all my comments, this review article only need an extensive linguistic review of all its parts.

Author Response

Point 1: After revising the entire review article and matching the authors' response to my previous comments on it, I believe that although the authors did not respond to all my comments, this review article only need an extensive linguistic review of all its parts.

Response 1: The text has been severely checked throughout.